# Emojis Are Comprehended Better than Facial Expressions, by Male Participants

**DOI:** 10.3390/bs13030278

**Published:** 2023-03-22

**Authors:** Linda Dalle Nogare, Alice Cerri, Alice Mado Proverbio

**Affiliations:** Department of Psychology, University of Milano-Bicocca, Piazza dell’Ateneo Nuovo 1, 20162 Milan, Italy

**Keywords:** visual perception, emotion, emoji, emoticon, sex differences, anger, fear, emotional communication, texting

## Abstract

Emojis are colorful ideograms resembling stylized faces commonly used for expressing emotions in instant messaging, on social network sites, and in email communication. Notwithstanding their increasing and pervasive use in electronic communication, they are not much investigated in terms of their psychological properties and communicative efficacy. Here, we presented 112 different human facial expressions and emojis (expressing neutrality, happiness, surprise, sadness, anger, fear, and disgust) to a group of 96 female and male university students engaged in the recognition of their emotional meaning. Analyses of variance showed that male participants were significantly better than female participants at recognizing emojis (especially negative ones) while the latter were better than male participants at recognizing human facial expressions. Quite interestingly, male participants were better at recognizing emojis than human facial expressions per se. These findings are in line with more recent evidence suggesting that male individuals may be more competent and inclined to use emojis to express their emotions in messaging (especially sarcasm, teasing, and love) than previously thought. Finally, the data indicate that emojis are less ambiguous than facial expressions (except for neutral and surprise emotions), possibly because of the limited number of fine-grained details and the lack of morphological features conveying facial identity.

## 1. Introduction

The main goal of the study was to compare, in the two sexes, the ability to comprehend emotional states conveyed by real faces with those conveyed by emojis, which are symbols used in long-distance communication. Emojis are colored ideographs, somewhat larger than alphabetical characters, which represent smileys or other entities such as hand signs, animals, food, etc., and are typically used in computer- or smartphone-mediated communication, particularly in messaging systems and social networks. They are often complementary to the written text, and aimed at replacing non-verbal communication as much as possible [1]. Therefore, emojis seem to compensate quite effectively for the lack of non-verbal clues (such as body language, facial expressions, and voice prosody) in textual communication. In particular, face pictographs are used to enrich communication emotionally, promoting the expression of emotional states and feelings. Some studies, along with the extreme pervasiveness of this modern communication system in messaging, have demonstrated their effectiveness. For example, Aluja and coauthors [2] measured the acoustic startle reflex modulation during the observation of emoji emotional faces and found higher acoustically evoked startle responses when viewing unpleasant emojis and lower responses for pleasant ones, similarly to those obtained with real human faces as the emotional stimuli. Again, Cherbonnier and Michinov [3] compared emotion recognition mediated by different types of facial stimuli—pictures of real faces, emojis, and face drawings (emoticons, also known as smileys)—and found that, contrary to what one might suppose, the participants were more accurate in detecting emotions from emojis than real faces and emoticons. Liao et al. [4] (2021), comparing ERPs generated by viewing painful and neutral faces and emojis, found that P300 was of greater amplitude in response to faces than emojis, but both types of stimuli elicited late positive potentials (LPP) of a greater amplitude in response to negative than positive emotions. The authors concluded that emojis and faces were processed similarly, especially at the later cognitive stages.

Emojis and emoticons have become a worldwide means of expressing emotions in computer-mediated communication, but the neuroscientific knowledge regarding how the human brain recognizes them is still scarce. Not much is known, especially, about a possible sex difference in the ability to comprehend emojis. Some studies were performed to compare face and emoticon processing. The results seem to suggest a partial independence in the subserving neural circuits. Yuasa and coworkers [5] compared the cerebral activations during the observation of Japanese faces and Japanese emoticons (e.g., (^_^), combinations of vertically organized letters and symbols) through fMRI and found that the face fusiform area (FFA) was active only during the processing of human faces (as expected, e.g., [6]), and not of emoticons. However, the right inferior frontal gyrus, involved in the assessment of the stimulus emotional valence, was active during the processing of both types of stimuli. This was interpreted as an indication that, even if emoticons were not perceived as faces, they activated higher-order social areas devoted to the interpretation of face emotional meaning. Similarly, using emoji stimuli, Chatzichristos and coworkers [7] failed to find an FFA activation during the perception and matching of positive and negative emojis with words triggering autobiographical memories. However, they found the involvement of the emotional and social areas such as the right temporal and inferior frontal areas and the amygdala nuclei. Other neuroimaging investigations reported how FFA and the *occipital face area* (OFA) were strongly active during the perception of emoticons. This would explain how they are emotionally recognized and identified as facial stimuli [8,9,10].

The purpose of this study was manifold. First, the communicative effectiveness of sets of emoji and face stimuli was assessed through preliminary validation. Then, it was investigated to which extent the observers grasped the emotional meaning of real facial expressions and emojis, as a function of the stimulus type, valence, and complexity. Another important aim of the study was to test the existence of sex differences in the ability to recognize the affective content of faces vs. emojis. Numerous studies have provided evidence that female participants do better than male participants in tasks involving deciphering emotions through facial expressions and non-verbal communication cues [11,12,13,14,15,16,17]. Female individuals would also be more likely than males to express their emotional experiences to others [18]. This sex difference would be more pronounced with subtle emotions (such as sadness or fear); it would also reflect a greater interest of females in social information [12] and a more empathic attitude toward suffering of others [19]. In detail, some evidence suggest that females would be more expressive and prone to externalizing emotions than male individuals, from adolescence through adulthood [20]. In particular, females would more frequently express states of sadness, fear, shame, or guilt than male individuals, while males would exhibit more aggressive behavior than females when feeling angry [21,22].

On the other hand, some authors have speculated that a gender difference might also exist in emoji use. Evidence has been provided of a more frequent female use of emojis than male [23]. Consistently, higher emoji usage and familiarity ratings for female participants than for male participants were reported in a large American survey [24] assessing the valence and the familiarity of 70 smileys (facial emojis). Given that emojis were introduced into text-based communication to more effectively express emotions and describe affective information, it might be hypothesized that females would manifest more intensive use of emojis and emoticons than males, analogously to what occurs in real-life social interactions [15,25]. However, this does not seem to be the case. Chen and coworkers [26] documented that 7.02% of male users used at least one emoji in their typical messages while 7.96% of female were likely to use one or more emojis. Herring and Dainas [27] reported that female and male social media users similarly interpreted and used emojis in Facebook messages. However, the authors reported that male users were significantly more inclined to use sentiment-related emojis (heartlets) than female users were (male: 19.4%; female: 17.6%), which contradicts the psychosocial literature according to which males would be less willing to express love in real social interactions than females [28]. According to the authors, such a finding would imply that, although males are reserved when expressing their love in real life, they are more willing to express love through emojis in textual communication. After all, it is possible that social behaviors change as technological contexts change.

In order to assess the existence of possible sex differences in the way emotions are processed and recognized in real faces and in emojis, two different samples of university students, matched for sex, age, cultural status, and residence area, were presented with a large set of real faces and emojis and asked to rate them for comprehensibility. The degree of recognizability of human facial expressions relative to eight different human identities (half male, half female) was compared with that of emojis printed in eight different graphic styles. Seven different types of neutral, positive, and negative facial expressions were considered. Before experimentation, pictures of the stimuli were validated by different groups of students (of similar age and sex) recruited in the same university, by measuring the rates of correct recognitions.

On the basis of the previous literature, it was expected that female participants would outperform male participants in the recognition of empathy-related, subtle, human facial expressions (e.g., fear) [29]. In addition, it was generally hypothesized that the emoji and face recognizability scores would be roughly comparable, with some occasional advantage for emojis, depending on the specific emotion [30,31]. The artificiality of this symbolic system would, in fact, be counterbalanced by the emojis’ lack of ambiguity, limited number of fine-grained features, and lack of internal noise related to the variety of human identities.

For example, previous studies reported how fearful emojis were recognized better than real fearful expressions [30]. Furthermore, fearful faces tended to be recognized more slowly than other emotions [32,33]. This disadvantage might be due to the fear configural subtleness, or to the presence of details shared with the surprise expression (the wide-open mouth and eyes) in real faces.

On the contrary, emoji expressions would be more markedly distinctive across categories (e.g., fear emojis feature dark open mouths and evident white sclera in the eyes). Therefore, the emojis’ advantage over fearful human faces reported by some studies might be due to their more distinctive featural characteristics. 

## 2. Materials and Methods

### 2.1. Participants

Ninety-six healthy students of local university (48 males and 48 females) aged 18 to 35 years (mean age = 23.89) participated in the main study (preceded by a validation performed on a different sample). They were randomly assigned to the facial expression study (48 subjects—24 males, aged: 23.37; 24 females, aged: 23.37 years) or the emoji recognition study (48 subjects—24 males aged 23.62; 24 females, aged: 24.91). The current sample size was tested for power analysis by using the program G*Power3.1 for comparing two independent groups with alpha level = 0.05. They were all right-handed, as assessed through the administration of the *Edinburgh Handedness Inventory*. They all declared to have never suffered from psychiatric or neurological deficits and to have good or corrected-to-good vision. Before taking part in the experiment, participants signed written informed consent forms. The study was carried out in accordance with the relevant guidelines and regulations and was approved by the ethics committee of University of Milano-Bicocca (CRIP, protocol number RM-2021-401). It was conducted online and programmed via Google Forms, https://www.google.com/forms (accessed on 20 May 2021).

### 2.2. Stimuli

For stimuli, 56 different human facial expressions and 56 different emoji pictures (i.e., 112 different visual stimuli), depicting 8 variants of 7 universal facial expressions were used in this study. Eight different identities (4 female, 4 male) were used for human faces, while eight different emoji styles were used for emoji testing. Stimuli were created and validated as described below.:

Emoji. Emoji pictures were drawn from free web platforms (Apple, Google, Microsoft, WhatsApp, Twitter, Facebook, Joypixels, and www.icons8.it (accessed on 1 April 2021) and represented the six basic Ekman’s emotions [34]: happiness, sadness, surprise, fear, anger, and disgust, plus neutrality, in eight different styles. They were matched for size (4 cm in diameter) and average luminance. The original stimuli were slightly modified to adjust their average luminance, color saturation, and size. Figure 1 illustrates some examples of stimuli. Stimuli were preliminarily validated in a behavioral study by means of a test administered via Google Forms to select the most easily recognizable emojis among a much larger set of 168 emojis comprising 24 different emoji styles for 7 facial expressions.

Validation was previously performed on group of 48 students (24 males, 24 females) aged on average 23 years. Participants in stimulus validation had normal or corrected-to-normal vision, and never suffered from neurological or psychiatric disorders. They were shown randomly mixed sequences of stimuli, one at a time, and at the center of the screen. The task consisted of deciding, as accurately and quickly as possible, which of the emotion words provided was more appropriate to describe the observed expressions, by clicking a check mark. The eight sets (featuring 56 emojis) associated with the highest accuracy in performance were selected as experimental stimuli. Average hit rate for the final set was 79.40%. In more detail, accuracy was 94.53% (SD = 0.51) for happiness, 76.04% (SD = 1.85) for surprise, 84.12% (SD = 1.76) for sadness, 96.09% (SD = 2.1) for anger, 57.29% (5.04) for fear, 91.15% (SD = 0.99) for disgust, and 56.51% (SD = 0.99) for neutrality.

Faces. Ten Caucasian students (5 males, 5 females) aged about 23 years were recruited for photo shooting. They were required to not wear any paraphernalia (e.g., earrings, glasses, make up, bows, clips, necklaces, tattoos, and piercings) while mustaches or beard were not permitted. All were required to wear a black shirt and to gather their hair behind the ears. The dark area above the forehead generated by the hairline was kept to preserve the humanness of the faces (the hair being, moreover, barely visible). For each of the seven emotional expressions, the actors had to portray a specific emotional state, by focusing on a specific autobiographical episode through the Stanislavsky method, and express spontaneously their mood. This procedure induced actors to activate their emotional memory, by recalling specific past experiences, and to react to them by externalizing spontaneous emotions, instead of concentrating on reproducing facial mimics dissociated from the momentarily emotional state, which might look phony. They were recommended to activate a positive scenario for ‘surprise’ (see Figure 1 for examples of stimuli). All participants provided written informed consent and signed the privacy release form.

Stimulus set was validated on a sample of 50 students (25 females, 24 males, and 1 genderfluid), aged about 23.7 years and different from the experimental sample. Participants in stimulus validation had normal or corrected-to-normal vision, and never suffered from neurological or psychiatric disorders. They were shown randomly mixed sequences of stimuli, one at a time, and at the center of the screen. The task consisted of deciding, as accurately and quickly as possible, which of the emotion words provided was more appropriate to describe the observed expression, by clicking a check mark. Stimuli were 56 pictures depicting the seven facial expressions exhibited by the 8 different actors. The results showed a high accuracy in correctly categorizing facial expressions (87.35%, in line with Carbon [35]); namely, hits were 98.5% for happiness, 86.7% for surprise, 80.1% for sadness, 89.3% for anger, 72.7% for fear, 85.97% for disgust, and 98.2% for neutrality. Stimulus set was also rated for facial attractiveness by a further group of 12 students (7 females and 5 males) aged 18–25 years. They were asked to rate the attractiveness of a neutral expression belonging to the 8 identities, by using a 3-point Likert scale (1 = “not attractive”, 2 = “average”, and 3 = “attractive”). The results showed no difference in attractiveness between individuals of the two sexes, with an average score of 1.83 for females and 1.82 for males. Overall actors were rated as “average” for attractiveness, which allows generalizability of results to the normal-looking population. Face stimuli were 3.37 cm × 5 cm (3°22′ × 5°) in size.

### 2.3. Procedure

The emotion-recognition task consisted of 112 experimental trials, in which participants were first shown a portrait photograph of an adult face (or a facial emoji, according to the experiment) to be inspected for about 2 s. The stimuli were equiluminant as shown by an ANOVA performed on luminance values (F = 0.1, *p* = 0.99). Photos of faces and emoji were in color, and were displayed at the center of the screen, on a white background. Immediately below the face or the emoji, there was a list of words (neutrality, happiness, surprise, fear, anger, sadness, and disgust), from which they had to select the emotion that they deemed the most appropriate to describe the meaning of the expression. Moreover, participants rated the degree of recognizability of the expression on a 3-point Likert scale. (1 = ‘not much’, 2 = ‘fairly’, and 3 = ‘very much’). The emotion was scored 0 if a different incorrect expression was selected. The time allowed for perceiving and responding to the two queries was 5 s. Participants were instructed to observe one facial stimulus at a time and to respond within 5 s, not missing any answer. Only one choice per face/emoji was allowed. The task lasted about 15 min.

### 2.4. Data Analysis

The individual scores obtained from each individual, for each of the 7 facial expressions and stimulation condition, underwent a 3-way repeated-measure ANOVA whose factors of variability were: 1 between-group factor named “sex” (with 2 levels: female and male), 1 between-group factor named “stimulus type” (with 2 levels: emoji and face), and “emotion” between-group factor (with 7 levels: happiness, neutrality, surprise, anger, sadness, fear, and disgust). Multiple post-hoc comparisons were performed using Tukey’s test. Greenhouse–Geisser correction was applied in case of epsilon < 1 and epsilon corrected *p* value were computed. Finally, a two-way ANOVA was also applied to all the raw data to measure the distribution of individual scores in relation to the sex of viewer and type of stimulus (regardless of facial expression).

## 3. Results

The ANOVA performed on the recognition rates of emojis vs. faces yielded the significance of Emotion (F 6,552 = 31.575, *p* < 0.00001). Post-hoc comparisons showed that happiness was considered the most recognizable emotion (2.639, SE = 0.028) and fear the least recognizable emotion (1.86, SE = 0.07), differing from all the others. The significant interaction of the Emotion x Stimulus type (F 6,552 = 13.87, *p* ˂ 0.00001) and relative post-hoc comparisons showed that the recognizability rates were higher for emojis than faces, for all emotions except happiness and the neutral expressions (see Figure 2).

The significant interaction of the Emotion × Sex × Stimulus type (F 6,552 = 2.3, *p* ˂ 0.03), and post-hoc tests, showed that female participants generally outperformed male participants in the recognition of facial expressions (especially anger, surprise, sadness, and disgust), while male participants outperformed female participants in the recognition of all emojis except fear (see Figure 3 for means and standard deviations).

Overall, while female participants were better at recognizing facial expressions than male participants, especially for surprise (*p* < 0.04), anger (*p* < 0.01), and sadness (*p* < 0.01), male participants were better at recognizing all emojis (especially neutral, surprise, and disgust; *p* < 0.01) than female participants, except for the emotion of fear, which was better recognized by females. 

The two-way ANOVA performed on the recognizability scores as a function of the viewer’s sex and stimulus type showed the significance of the stimulus (F 1,167 = 15.38, *p* < 0.0002), with higher scores attributed to emojis than faces, and of the sex (F 1,167 = 40, *p* < 0.0001), with a better performance for female than male participants. However, the significant interaction of the Stimulus type × Sex (F 1,167 = 65, *p* < 0.0001) and relative post-hoc comparisons showed that, while there was no difference between emoji vs. faces for female participants, males performed much better than females in recognizing emojis (*p* < 0.0033), and much worse than females in recognizing facial expressions (*p* < 0.00008). Furthermore, males were much better at recognizing emojis than facial expressions (*p* < 0.00008), as clearly visible in Figure 4.

## 4. Discussion

This study compared the recognizability of human facial expressions and emojis, balanced by number of emotions and number of styles/identities tested. Regardless of the sex of the viewers, some emojis were recognized more clearly than facial expressions, especially anger. This result is similar to that found by Fischer and Herbert [30], contrasting facial expressions, emoji, and emoticons, and finding that emojis and faces were quite good at representing the associated emotions and therefore also in reducing ambiguity. 

In addition, for some emotions (i.e., anger), emojis were even better than faces. It should be considered that angry and disgusted emojis, for example, have a slightly different coloring (reddish and greenish), which makes them distinctive. Quite similarly, Cherbonnier and Michinov [3], comparing the comprehension of real faces, emojis, and face drawings, found that subjects were more accurate in detecting emotions from emojis than from the other stimuli, including faces, especially for negative emotions such as disgust and fear.

On the whole, in this study, happy emojis were recognized more accurately than the other emojis, while fear expressions were recognized more inaccurately. Overall, in this study, happy emojis were recognized more accurately than others were, while fearful expressions were recognized with greater uncertainty. This pattern of results fits with the findings reported by some authors [31,32], and showing faster RTs to joyful expressions and slowest for fearful expressions. Similarly, Fischer and Herbert [30] found the fastest responses for happiness and anger, followed by surprise, sadness, neutral, and lastly, fear. The same pattern of results was reported for real human faces [36]. 

These data are also in accordance with previous research literature showing how children were more accurate in recognizing happy and sad emoticons, while doing worse in recognizing fear and disgust emoticons [23]. The primacy of the expression of happiness might be linked to its specific and unique physiognomic characteristics (e.g., the typical U-shaped mouth curvature) not visible on other emotional expressions [36], or to the fact that positive expressions are more commonly exhibited in social settings (e.g., social smiling), as hypothesized by Schindler and Bublatzky [37]. A similar advantage for recognizing happiness has been found with the Emoji stimuli, also characterized by the (even more prominent) U-shaped mouth feature for smiling. The emotion of fear, on the other hand, was often confused with other emotions, particularly with surprise, as both are characterized by the arching of eyebrows and the widening of the eyes and mouth.

In this study, female participants outperformed male participants in the recognition of fearful emojis. This could be partly related to the fact that fearful emojis were more obscure and difficult to be distinguished from surprise, as the hits were only 57.29% in the validation assessment. Similarly, in a study by Hoffmann et al. [38], it was found that female participants were significantly better at recognizing anger, disgust, and fear (subtle emotions) than male participants. According to the authors, this finding was related to the greater ability of females to perceive emotions in a gestalt fashion, making quick and automatic judgements (see also [39] in this regard). Again, it was found that females were faster than males at recognizing fear, sadness, and anger: they were also more accurate than males in recognizing fear [40]. In this study, statistical analyses showed that male participants were significantly better than female participants at recognizing emojis while females were better than males at recognizing human facial expressions. Quite interestingly, males were better at recognizing emojis than human facial expressions. In general, the female ability to recognize human facial expressions more accurately (or faster) than males is a well-corroborated notion in psychology and cognitive neuroscience [15,41]. Further evidence has demonstrated that female individuals, compared to male individuals, react more strongly when viewing affective stimuli (such as *International Affective Picture System* images) involving human beings, thus showing a higher empathic response [42,43,44]. Neuroimaging evidence has also been provided on the fact that face processing would engage FFA bilaterally in females, and unilaterally (i.e., only over the right hemisphere, rFFA) in males [45], thus possibly supporting a deeper/more efficient analysis of subtle facial mimicry in females. However, the presumed female superiority in the ability to comprehend emotional non-verbal cues does not seem to include the processing of emojis. Therefore, it is possible that emojis were processed differently, half-way between symbols (such as emoticons or icons) and real faces. Indeed, emojis are colored ideographs, not real body parts. While possessing some facial features able to stimulate social brain areas (e.g., the OFA, the middle temporal gyrus, and the orbitofrontal cortex), emojis seem to stimulate weakly, or not at all, the face fusiform area, sensitive to configural face information [7].

An interesting study by Wolf [46] observing the effect of gender in the use of emoticons in cyberspace found that, to communicate with women in mixed-gender newsgroups, men adopt the female standard of expressing more emotions, being, therefore, not less expressive, as with real facial mimicry. The majority of emoticon use by women would however lie in the meaning category of humor, and include solidarity, support, assertion of positive feelings, and thanks, whereas the bulk of male emoticon use would express teasing and sarcasm. This gender difference in the used emoji valence is compatible with the present pattern of results, showing how male participants outperformed female participants with negative emojis while the latter were better than the former at deciphering positive emojis such as happy or even neutral ones.

Overall, it can be hypothesized that the unexpected male proficiency in emoji recognition might reflect a progressively greater inclination of male participants (namely, students) to express their emotions, particularly negative, but also amorous ones (according to other studies), through messaging in cyberspace. In addition, it may be enlightening to compare the present pattern of results with the hyper-systemizing theory of Baron-Cohen [47] according to which males would be better at processing visuospatial information at the analytic level (e.g., emojis or emoticons), and women better at empathizing with others (e.g., on fearful or sad facial expressions). These changes in social behavior should be further assessed and investigated over time. 

It should be considered, however, that no empirical data directly suggest a male superiority in the ability to visually process or recognize objects, symbols, icons, or alphanumeric characters. No sex difference was ever shown in the ability to process object identity [48], visually recognize pictures of detailed objects [49], or visually recognize words [50].

On the other hand, several pieces of evidence suggested an association between the male sex, as well as high testosterone levels, and a better performance in visuo-spatial tasks, including the rotation of 3D shapes [51,52]. Although some authors have found a correlation between ‘systemizing’ male traits and performance in mental rotation tasks, e.g., [53], the visuo/spatial or spatial rotation ability does not seem to be correlated with face processing, or visual processing of colored face-like objects. It can ultimately be hypothesized that males in this study were less attracted to, or attentively oriented toward, social information [12,15], represented in this study by real human faces, and therefore were less able to capture subtle facial mimics (such as that of the surprise, anger, and sadness expressions). Sexual dimorphism in attention to social vs. non-social (object) stimuli has been widely reported [54,55] and might ultimately contribute to explaining why male participants in this study (unlike females) were worse at reading emotions in human faces than in emojis.

## 5. Study Limits and Further Directions

It should be interesting to explore a wider range of emojis to further investigate this sex effect. It would also be important to gain neuroimaging data to explore if these differences across sexes are paralleled by sex differences in the activation of visual areas devoted to face vs. emoji processing. One limit of the present study is the lack of response time data, due to the methodological paradigm involving qualitative and quantitative assessment but not response speed measurement (because of COVID-19 restrictions). The other limit that should be considered is that, here, facial expressions were shown as static pictures and not videos, which can make the recognition of dynamic facial expressions more difficult.

## Figures and Tables

**Figure 1 behavsci-13-00278-f001:**
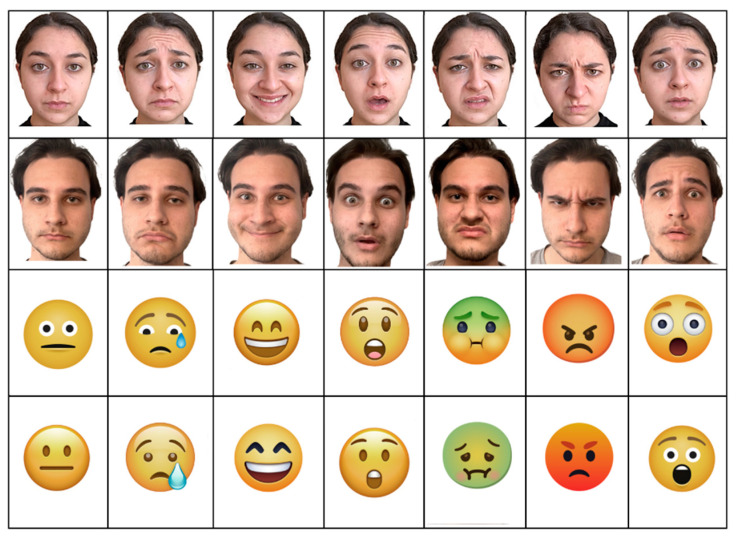
Examples of facial expressions and emojis used to illustrate the 7 affective states, in 4 different “identities”: 1 female, 1 male, and 2 different emoji styles.

**Figure 2 behavsci-13-00278-f002:**
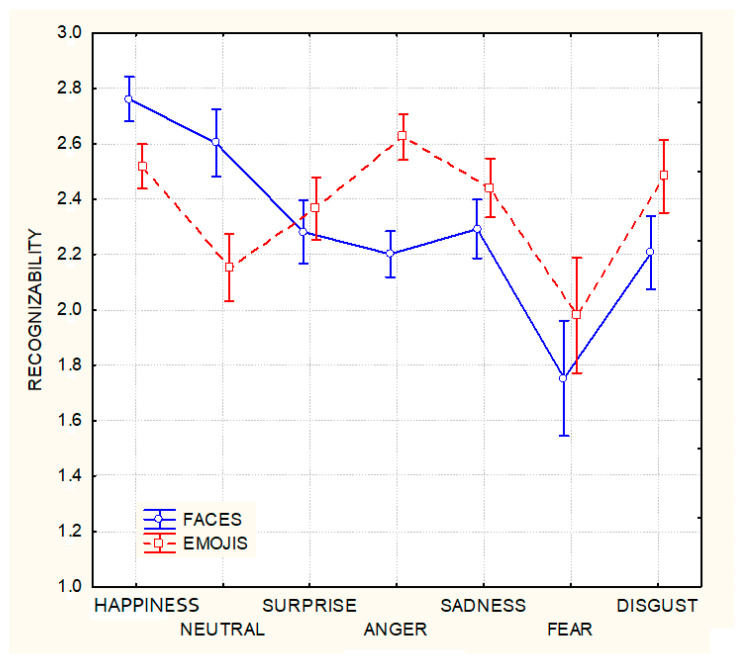
Mean recognizability scores (with standard deviations) relative to the 7 expressions illustrated by human faces and emoji.

**Figure 3 behavsci-13-00278-f003:**
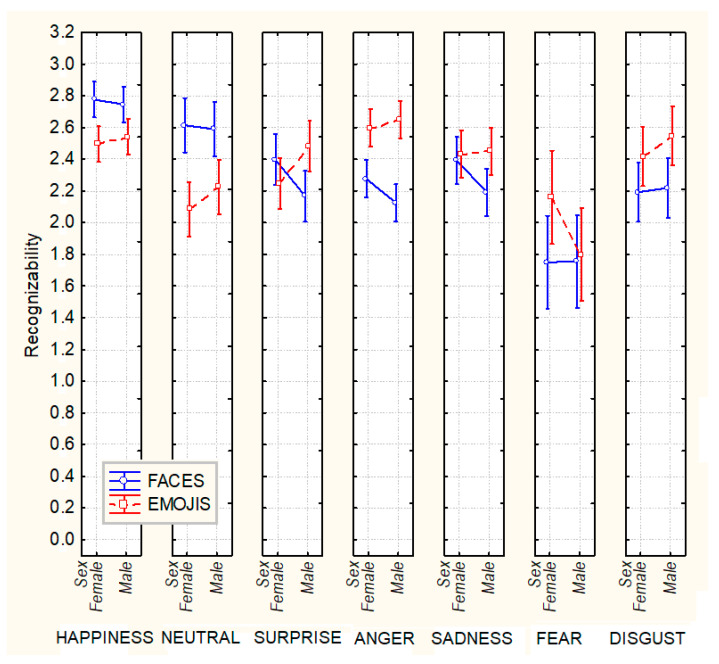
Mean recognizability scores (with standard deviations) measured in male and female participants in response to the seven emotions expressed by human faces and emojis.

**Figure 4 behavsci-13-00278-f004:**
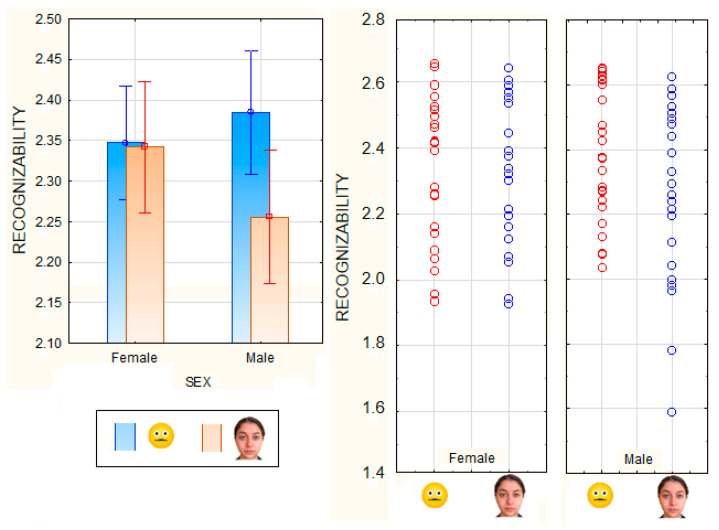
Mean scores and distribution of individual performance as a function of stimulus type and sex of viewers. On the y-axis are visible recognizability scores, while on the x-axis are subjects’ sex and stimulus.

## Data Availability

Anonymized data and details about the preprocessing/analyses are available to colleagues upon request.

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
