# Peer review of "Emojis Are Comprehended Better than Facial Expressions, by Male Participants"

_behavsci, 2023, doi:10.3390/bs13030278_

Round 1

Reviewer 1 Report

The research topic fits in with the current trend, and the thesis has the practical value or innovative findings. There are several suggestions as below. If you could make proper modifications, I believe it could better reveal the value of the thesis.

1. Introduction shall make the reader further understand the research. It is suggested to clearly point out the research purpose and problems trying to be solved, and express the author’s standpoint in the end, to set the tone of the thesis.

2. There shall have precise variable control and clear steps in the research method, to effectively promote the reliability and validity of the research result.

3. The quality of presentation and result explanation shall be improved for the data evidence and data analysis.

4. For the Conclusion part, it is suggested to increase more specific academic connotation and practical description, to reveal the thesis value and influence better. 

Reviewer 2 Report

Although the subject matter is very interesting and the article is well structured, I cannot support this article and therefore reject it for the following reasons: 

1.- When submitting the file to the anti-plagiarism programme (Turnitin) it shows a similarity index of 53% (attached file).

2.- Source 1 with 19% similarity, corresponds to the publication of the following URL: https://www.frontiersin.org/articles/10.3389/fnins.2022.864490/full

3.- Source 2 with a 15% similarity, is at www.researchgate.net

Reviewer 3 Report

In general the study is clear and well conducted. However there is a problem related to the description of the different experimental phases and participants which performed each phase.

If I have correctly understood the study consists of a first "validation" phase in which authors study the ability of two groups of participants ( 48 and 50, respectively) of recognizing emoji and facial espression valence.Then a group of, maybe 48, participants performed a task in which they were asked to recognize the valence of 112 stimuli ( 56 emoji and 56 faces) .

Thus , if my interpretation of the protocol is correct, I don't  understand while at the beginning of the Participants session the authors write: "96 healthy students of local University (48 males and 48 females) aged 18 to 35 years (mean age = 23.89) participated in the study. They were randomly assigned to the facial expression study (48 subjects: 24 males, aged: 23.37; 24 females: 23.37 years) or the emoji recognition study (48 subjects: 24 males aged 23.62; 24 females: 24.91)."

Is the "facial expression study" the validation study in which the authors investigated the ability to recognize both emoji and facial expressions?

And is the "emoji recognition study " the other session which however involves the use not only of emoji but also of faces?

Moreover, also the number of the participants is not clear.

I think the authors shoud rewrite this section in order to make it clearer to the reader.

Reviewer 4 Report

I have read the paper titled “Are men better at recognizing emojis than facial expressions?” by Delle Nogare and colleagues. The paper is interesting and could be useful in deepen the knowledge in the specific field of visual perception of emotion. Unfortunately, the proposed paper has many flaws and needs major revisions.

Title:

The paper title is a question. A question title should be used when authors think that are multiple perspectives to interpret the data. However, it seems to me that results are interpreted in a straight way and the conclusions are clear cut. A question title might be persuasive and should be avoided in this case.

Introduction

Authors have inconsistently used  the term male/female and men/women. It should be used male participants and female participants instead of “men” and “women”which are more colloquial terms. It should be noted that “man” and “woman” are gender related terms while “male” and “female” are terms relating to biological sex. If the paper is about sex differences I suggest to use “male participants” and “female participants” or at least to be consistent throughout the manuscript.  

-        They are effectively used in textual communication to compensate for the lack of nonverbal signs and to enrich communication emotionally, promoting the expression of emotions.”

It is not clear if the emotional enrichment is related only to emoji representing faces or also in case of other type of emoji. Please specify

-        Cherbonnier and Michinov [3] compared emotion recognition mediated by different types of facial stimuli: pictures of real faces, emojis, and face drawings and found that, contrary to what one might suppose, participants were more accurate in detecting emotions from emojis than real faces and emoticons

It isn’t clear if “emoticon” in the last sentence is referred to emoticon representing face or other kind of images, please specify.

-        “In addition, it was generally hypothesized that emoji and face recognizability scores would be roughly comparable, with some occasional advantage for emojis, depending on the specific emotion [25-26]. The artificiality of this symbolic system would in fact be counterbalanced by emojis’ lack of ambiguity, limited number of fine-grained features, and lack of internal noise related to the variety of human identities.”

Authors should explain the specific emotions which are better recognized by observing a emoji rather than a face. Is this phenomenon due to specific interplay between configural and featural characteristics of a seen stimulus (face or emoji)? 

-        Women would also be more likely than men to express their emotional experiences to others [15].

This statement is vague and quite generic, explain better the differences between male and female in emotion perception. Also, it is not clear the relationship between emotion perception and emotion expression. Authors should better explain the link between emotion perception and emotion expression

Authors might find some insight in https://pubmed.ncbi.nlm.nih.gov/23231534/

https://link.springer.com/article/10.1007/s10339-005-0050-6

https://www.sciencedirect.com/science/article/pii/S0301051111001384

https://www.tandfonline.com/doi/full/10.1080/02699931.2018.1454403?casa_token=HZcDDMKn8_kAAAAA%3A1QnJdkJWx9A8H1X-JZ4xRd2TozSj_btilW3XYN693tgtPjlR1nWItS8Ib5FBs37n1Smfqwwd6i-qMQ

https://www.sciencedirect.com/science/article/pii/S1053811916307376

Materials and methods

The term “joy” should be replaced by “happiness”

-        “For each of the seven emotions, actors were instructed to imagine a vivid emotional state, while concentrating on a specific autobiographic scenario through the Stanislavsky method, and express it spontaneously”

Authors should explain why they chose the Stanislavski method. Furthermore, in emotion perception research it is usual to use face from standardized database. Usually when doing research in the field of emotion perception it is preferable to use stimuli in which hair are not visible. In the present research authors have compared emoji which are composed by simple shape with stimuli such as faces which have external element (the hair). Could differences in face shape and external feature (the hair) partially explain the results from this study?

-        “in a 3-point Likert scale (ranging from ‘1 = not much’ to ‘3 = very much’).”

Since it is only a three point Likert scale authors should mention alle the labels associated to the numbers.

Analysis

-        It is not clear the choice of a Wilcoxon test and what this test adds to the previous ANOVA. Wilcoxon test is a non-parametric test, the choice of parametric and non parametric test should be based on the shape of data distribution.

-        Is figure 4 related of the Wilcoxon test? Please explain the meaning and what x and y scales represent in caption

Discussion

Discussion section should be improved by expanding the relationship between the results and the differences in emotion perception and visuospatial processing between male and female. Authors should take in consideration research about face perception and emotion perception and deepen the discussion.

-        “However, the presumed female superiority in the ability to comprehend emotional nonverbal cues does not seem to include emojis’ processing. Therefore, it is possible that emojis are processed differently, half-way between symbols (such as emoticons or icons) and real faces”

Please explain better the process behind the emoji perception and the meaning of half-way.

-        “In addition, it may be enlightening to compare the present pattern of results with the hyper-systemizing theory of Baron-Cohen [43] according to which males would be better at processing visuospatial information at analytic level (e.g., emojis or emoticons), and women better at empathizing with others (e.g., on fearful or sad facial expressions)”

This part of the limits paragraph might be expanded and discussed in the discussion section

Round 2

Reviewer 2 Report

The article has been improved. Please correct the references according to the ASC style, some are missing the doi, others are missing a space after a comma. Regards

Author Response

Dear Reviewer 2,

  1. References were corrected according to the ASC style,
  2. missing dois were added,
  3. missing spaces after a comma were added.

Thanks for your review

Reviewer 4 Report

Introduction, Line 5: Please use the term smartphone instead of iphone. 

Author Response

Dear reviewer,

the term smartphone has been used instead of iphone. Thanks